# Functional Role of Natriuretic Peptides in Risk Assessment and Prognosis of Patients with Mitral Regurgitation

**DOI:** 10.3390/jcm9051348

**Published:** 2020-05-05

**Authors:** Giovanna Gallo, Maurizio Forte, Rosita Stanzione, Maria Cotugno, Franca Bianchi, Simona Marchitti, Andrea Berni, Massimo Volpe, Speranza Rubattu

**Affiliations:** 1Department of Clinical and Molecular Medicine, School of Medicine and Psychology, Sapienza University of Rome, 00189 Rome, Italy; giovanna.gallo@uniroma1.it (G.G.); andrea.berni@uniroma1.it (A.B.); massimo.volpe@uniroma1.it (M.V.); 2IRCCS Neuromed, 86077 Pozzilli (Isernia), Italy; maurizio.forte@neuromed.it (M.F.); stanzione@neuromed.it (R.S.); maria.cotugno@neuromed.it (M.C.); franca.bianchi@neuromed.it (F.B.); simona.marchitti@neuromed.it (S.M.)

**Keywords:** natriuretic peptides, mitral valve regurgitation, valve repair, valve replacement, risk prediction

## Abstract

The management of mitral valve regurgitation (MR), a common valve disease, represents a challenge in clinical practice, since the indication for either surgical or percutaneous valve replacement or repair are guided by symptoms and by echocardiographic parameters which are not always feasible. In this complex scenario, the use of natriuretic peptide (NP) levels would serve as an additive diagnostic and prognostic tool. These biomarkers contribute to monitoring the progression of the valve disease, even before the development of hemodynamic consequences in a preclinical stage of myocardial damage. They may contribute to more accurate risk stratification by identifying patients who are more likely to experience death from cardiovascular causes, heart failure, and cardiac hospitalizations, thus requiring surgical management rather than a conservative approach. This article provides a comprehensive overview of the available evidence on the role of NPs in the management, risk evaluation, and prognostic assessment of patients with MR both before and after surgical or percutaneous valve repair. Despite largely positive evidence, a series of controversial findings exist on this relevant topic. Recent clinical trials failed to assess the role of NPs following the interventional procedure. Future larger studies are required to enable the introduction of NP levels into the guidelines for the management of MR.

## 1. Introduction

Mitral regurgitation (MR) represents one of the most frequent valve diseases with an indication for valve replacement or repair, both as surgical or transcatheter interventional management [1].

The etiology of mitral dysfunction, namely primary or secondary regurgitation, should be clearly identified. In primary or organic MR, the valve apparatus is directly affected as a consequence of a degenerative (i.e., fail leaflet or prolapse) or infective (i.e., endocarditis) process. In secondary or functional MR, the structure of the components of the valve apparatus, such as leaflets and chordae, is preserved, but an impaired left ventricular (LV) geometry is responsible of an altered balance between closing and tethering forces on the valve. In both abovementioned conditions, MR is responsible for or contributes to the development of LV and left atrial (LA) overload, leading to hemodynamic alterations.

According to the most recent European Guidelines [2], urgent surgery is recommended in cases of acute severe MR. In chronic primary MR, valve replacement or repair is indicated in symptomatic patients and, in the absence of symptoms, in the presence of LV ejection fraction (LVEF) <60%, LV end-systolic diameter (LVESD) ≥45 mm, atrial fibrillation (AF), and systolic pulmonary pressure ≥50 mmHg. Valve repair should be preferred if feasible and able to achieve a durable result with a low risk of re-intervention, such as in segmental valve prolapse. Rheumatic lesions, leaflets, or extensive annular calcifications more often require valvular replacement, preferably preserving sub-valvular apparatuses. In patients with high surgical risk, percutaneous edge-to-edge mitral repair is currently widely adopted and recommended by international Guidelines [2,3].

In secondary MR, due to significant operative mortality, high rates of recurrent MR, and the absence of a definite survival benefit, surgery is indicated when concomitant coronary revascularization is required. Even in this circumstance, percutaneous edge-to-edge repair may represent an efficacious option [4,5].

Although a “watchful waiting” strategy is considered safe and is accepted in asymptomatic patients, the assessment of correct and univocal timing of surgeries still remains a challenge. The identification of symptoms may be difficult due to its subjective nature and to the risk that patients could minimize their clinical manifestations in order to delay surgery, or could progressively reduce their activities as a consequence of an impaired functional capacity. In addition, symptoms may become clear when LV dysfunction is irreversible [6].

In this complex scenario, one of the unmet needs is a more accurate risk stratification, in which biomarkers may represent a useful tool to identify patients with a possibly unfavorable prognosis under conservative management or after mitral valve (MV) surgery.

Among several biomarkers available in clinical practice, natriuretic peptides (NPs) have a well-established role in cardiovascular diseases and, particularly, in heart failure (HF) where they reflect cardiac overload, LV systolic and diastolic dysfunctions, and are associated to cardiovascular outcomes [7].

Atrial natriuretic peptide (ANP), brain natriuretic peptide (BNP), and their inactive *N*-terminal portions (NT-proANP and NT-proBNP) are released in response to increased myocardial stretch, as a consequence of volume or pressure overload. NPs exert several cardiovascular and renal actions mediated by the type A natriuretic peptide receptor through the second messenger cGMP. The effects of NPs are able to counterbalance hemodynamic congestion through the regulation of electrolytes, water balance, and permeability of systemic vasculature. Moreover, they inhibit the renin–angiotensin–aldosterone system and the sympathetic nervous system. At the cellular level, a modulatory role on cellular growth and proliferation is recognized [8,9,10]. Some differences exist among NPs. ANP is stored in granules as a previously synthesized pool within the atrial cardiomyocytes and is quickly released upon request. BNP production is regulated by gene expression in ventricular cardiomyocytes, secreted as a prohormone, then cleaved into the active peptide and the NT-proBNP. As compared to ANP, BNP has a longer half-life (1–2 hours compared to approximately 22 minutes) and greater plasma concentrations (about 10-fold higher) [11,12]. Moreover, it has been proposed that NPs’ responses may depend on the specific pathophysiology of the underlying cardiac stress, suggesting that ANP may be more sensitive in the case of subclinical damage, whereas the BNP level shows a greater increase in acute conditions [13,14].

The aim of our review is to analyze the current available evidence on the role of NPs in the management, risk evaluation, and prognostic assessment of patients with MR before and after surgical or percutaneous valve repair.

## 2. NPs and Risk Assessment in MR

Several studies have investigated the association between increased levels of NPs in patients with MR, parameters of LV dysfunction, and cardiovascular outcomes.

An analysis of data obtained in 1399 patients from 15 studies found a positive relationship between levels of NPs and LV end-systolic parameters, such as the LV end-systolic index (LVESI) and LVESD [15]. BNP level was also associated with the myocardial performance index (MPI), an echocardiographic index of systolic and diastolic function [16]. Mayer and colleagues documented that patients with severe MR and BNP values >409 pg/mL had a mean LVESD of 40 mm [17], which represents a criterion for surgery according to U.S. Guidelines [3]. A linear relationship between the LVESD value of 40 mm and NT-proBNP level >292 pg/mL was identified by Potocki et al. [18]. An elevated BNP level has been documented in patients with pulmonary artery pressure >50 mmHg [18], this parameter being another indication for valve replacement or repair [2,3]. NT-proBNP values were directly related to HF functional classes in MR, with mean levels of 97 pg/mL for New York Heart Association (NYHA) class I, 170 pg/mL for class II, and 458 pg/mL for class III [19]. The addition of BNP to the Society of Thoracic Surgeons (STS) score improved the risk stratification in patients with primary MR and preserved LVEF [20].

In a prospective study conducted in 124 patients with chronic primary MR, Detaint and colleagues analyzed the relationship between BNP level, MR degree, LV and LA remodeling, and prognosis [21]. BNP level was associated to the LV end-systolic volume index, LA volume, and symptoms, and it was able to predict prognosis independently from age, sex, functional class, MR severity, and LVEF. At the 5-year follow-up, survival was significantly worse in patients with BNP level >31 pg/mL, showing a higher incidence of the combined end-point of death and HF [21].

In another study, BNP level >105 pg/mL had stronger predictive power compared to the most common parameters of MR severity, such as an effective orifice regurgitant area (EROA) and LVESD [22]. In 49 patients with MR and preserved ejection fraction (>55%), BNP > 41 pg/mL and NT-proBNP > 173 pg/mL showed the best accuracy in predicting the development of symptoms [23]. In 87 patients with severe MR, BNP below 80 pg/mL and NT-proBNP lower than 200 pg/mL showed the greatest negative predictive value of 98% for the development of symptoms or LV dysfunction during follow-up [24].

The prognostic role of BNP level in the management of MR has also been examined during exercise.

Exercise BNP level was strongly correlated with those measured at rest [25]. Patients with higher BNP values showed more severe MR, greater LA volume, more elevated systolic pulmonary pressure, and LV filling pressure estimated as an E/e’ ratio. Moreover, detected exercise BNP levels were higher in patients who developed symptoms and had an increased incidence of cardiac events, independently from age, gender, and body mass index. In patients with moderate MR, only the exercise LV global longitudinal strain, but neither resting nor exercise LVEF, was an independent determinant of exercise BNP level. A plausible explanation of these findings is that the BNP level has been assessed in patients with degenerative MR and not in MR secondary to LV dilatation, in which levels of NPs were measured before the development of LV systolic dysfunction [25]. A cut-off of 64 pg/mL was identified as the best cut-off value for exercise BNP level to behave as an independent predictor of worse cardiovascular outcomes [25]. Other studies have consistently demonstrated that an increase in BNP level during exercise is related to the development of HF, to subclinical LV dysfunction, and to reduced performance capacity [26,27].

According to the abovementioned results, BNP level may be used in clinical practice as a complementary tool for echocardiographic exams and exercise tests in order to identify those patients who are more likely to experience death from cardiovascular causes, and HF and cardiac hospitalizations, thus requiring surgical management, rather than a conservative approach. In addition, NPs may represent an essential tool to monitor the progression of the valve disease before the development of hemodynamic consequences in a preclinical stage of myocardial damage.

However, some controversial aspects deserve to be better-clarified. First of all, as documented by the abovementioned studies, it is still difficult to establish a univocal cut-off level able to identify MR patients at elevated cardiovascular risk, since different levels of NPs have been identified in the different studies. To overcome this problem, Clavel et al. introduced the BNP ratio, which is derived from a measured BNP value divided by the expected value related to the age and sex of each patient [26].

This parameter behaves as a significant independent prognostic marker of outcomes in valve heart disease patients, including MR patients under medical treatment [28,29]. However, it failed to maintain its role after surgery [28]. Interestingly, the BNP/ANP ratio revealed a prognostic role in one study, being significantly higher in the presence of clinical and echocardiographic criteria used for surgery recommendation, such as LVESD ≥ 45 mm, LVEF ≤ 60%, NYHA class II or greater, and AF [30].

Other critical issues relate to the appropriate time interval that should be considered for the subsequent measurements of the NP level, and the magnitude of changes of NP values between baseline and subsequent assessments, which may be related to a poor clinical outcome. In addition, age, AF, renal function, and body weight are known modulators of NP levels, thus representing potential confounders [31]. Finally, since a high BNP level may also be detected in patients with moderate MR, it cannot be used as a surrogate for MR quantification. On the other hand, it should be pointed out that, in the presence of good hemodynamic compensation with a normal NP level, regardless of LV dimension, the need for an interventional strategy may be missed.

## 3. The Role of NPs to Predict Outcome after MV Surgery or Percutaneous Repair

During the last few years, percutaneous edge-to-edge MV repair with the MitraClip (Abbott Vascular, CA USA) device has acquired increasing importance as a treatment option, especially in patients with HF with an elevated surgical risk. However, many patients are still being treated with the surgical MV repair or replacement, which represents the gold-standard procedure. Apart from the reduction of MR, both percutaneous and surgical interventions may produce several hemodynamic benefits, reducing LV and LA pressure and volume overload and, as a consequence, the myocardial wall stretch [2,3].

In this context, several studies have assessed the role of NPs in the management of patients treated with MV repair or replacement, investigating the sensibility of these biomarkers in identifying subjects with a worse response to the performed interventional procedure and with a lower chance of survival.

In a study involving 65 patients treated with edge-to-edge valve repair, a low NT-proBNP level, measured 6 months after the procedure, was associated with a significant reduction in LV end-diastolic volume (LVEDV) and end-systolic volume (LVESV) and to an improvement in LV and LA longitudinal strain [32]. Worse renal function, larger LVEDV, and a higher transmitral gradient after MitraClip (Abbott Vascular, CA USA) implantation were independently associated with a higher level of NT-proBNP at follow-up. Patients with low and medium NT-proBNP tertiles experienced a significantly greater reduction in NYHA functional class symptoms and quality of life score compared to those with a higher NT-proBNP level. At the 6-month follow-up, successful MR reduction was observed in a higher number of patients with low and medium NT-proBNP tertiles, whereas severe MR more often persisted in the high NT-proBNP tertile group (43%) [32].

After percutaneous valve repair, improvements in 6-minute walking distances and a decrease in LV volumes were paralleled by a significant reduction of NT-proBNP level [33].

Hwang and colleagues identified a BNP cut-off level of 125 pg/mL associated with a higher risk of cardiac death and re-hospitalization for cardiac causes in 117 patients who underwent surgical MV replacement [34]. Interestingly, a study conducted with 44 patients treated with transcatheter valve repair, as well as baseline levels of mid-regional proANP and NT-proBNP were significantly higher in those who experienced death or re-hospitalization for HF during a median follow-up of 211 days [35]. In a cohort of 174 retrospectively examined patients, the NT-proBNP level was significantly associated to survival at univariate analysis, but the independent predictive power of NT-proBNP was not confirmed at multivariate analysis [36]. However, the post-operative NT-proBNP level maintained its predictive role for a clinical outcome at multivariate analysis in other studies [37,38,39].

As for the pre-operative level, controversial findings were frequently reported with regard to the post-operative level. In 59 patients who underwent percutaneous MV repair, achieving a reduction of MR, an improvement of functional class, and structural reverse cardiac remodeling (reduced LA volume and LVESD and increased LVEF), the NT-proBNP level did not decrease significantly [40]. Similar findings were provided by Yoon et al. in a cohort of 144 patients successfully treated with edge-to-edge repair, in which the NT-proBNP level did not significantly decrease after MV clipping. In addition, NT-proBNP changes were not related to baseline LVEF and LV diameter and were not able to predict cardiovascular outcomes during a 6-month follow-up [41]. Furthermore, in a study which enrolled 194 patients treated with percutaneous valve repair, the NT-proBNP level remained elevated (≥10,000 pg/mL) in those patients who achieved a reduction of MR to grade ≤2 (21%) [42]. These controversial results may be explained by differences in the sample size of the enrolled populations, in the characteristics of the included patients (baseline LVEF, pulmonary artery pressure, LV diameter and diastolic function), in the duration of the follow-up, and also the rhythm status (i.e., sinus rhythm or AF) [43].

Finally, as for the pre-surgical management, it has not been clearly established which biomarker should be chosen, which cut-off value should be considered for the risk assessment of the patients, and which interval for serial monitoring of NPs has the best accuracy.

## 4. NPs Levels in MITRA-FR and COAPT Trials

Two recent trials have become available in patients subjected to MV repair. The MITRA-FR (Percutaneous Repair with the MitraClip Device (Abbott Vascular, CA USA) for Severe Functional/Secondary Mitral Regurgitation) trial, conducted in patients with severe secondary MR, showed that percutaneous MV repair added to standard pharmacological therapy was unable to reduce the rate of death or unplanned hospitalizations for HF at 1 year compared to those who received medical therapy alone [44].

The Cardiovascular Outcomes Assessment of the MitraClip Percutaneous Therapy for Heart Failure Patients with Functional Mitral Regurgitation (COAPT) study demonstrated that the transcatheter MV repair reduced the rate of hospitalizations for HF and all-cause mortality within 12 and 24 months of follow-up [45]. Although the MITRA-FR and COAPT trials enrolled comparable populations of patients with secondary MR, they obtained diametrically opposed results [44,45].

Several possible mechanisms have been proposed to explain the discrepant findings, mostly focusing on the different echocardiographic characteristics of the subjects included in the two trials [46,47]. The COAPT excluded patients with very severe LV dilation (LVESD < 70 mm), whereas LV diameter did not represent an exclusion criterion in MITRA-FR. This resulted in a significant difference in the documented mean LV volume (LVEDV 135 ± 35 mL/m^2^ in MITRA-FR vs. 101 ± 34 mL/m^2^ in COAPT) [44,45,46,47]. More interestingly, the two populations had a different degree of MR, the EROA being significantly greater in COAPT as compared with MITRA-FR (41 ± 15 mm^2^ vs. 31 ± 10 mm^2^, respectively) [44,45,46,47]. According to these considerations, it has been supposed that the underlying cardiac disease was probably the main cause of HF and the determinant of prognosis in MITRA-FR, with MR being a marker of adverse LV remodeling. In contrast, the LV dysfunction was more related to MR severity in COAPT, which also represented the main contributor to outcomes [48]. In such a context, it has been proposed that the degree of MR was “proportionate” to the degree of LV dilatation in MITRA-FR, whereas it was “disproportionate” in COAPT, and this parameter may have influenced the different clinical response to the percutaneous repair procedure [46,47].

In this complex scenario, the differences in the baseline NP levels between the two studies may mirror the different pathophysiological mechanisms. In the COAPT trial, both NT-proBNP and BNP levels were significantly higher than in MITRA-FR (median NT-proBNP > 5100 vs. 3200 pg/mL and median BNP >1000 vs. >760 pg/mL, respectively), and it may be argued that they were more related to MR severity than to either LVEF or LV diameter [44,45]. In fact, a higher degree of MR may have produced an increase in atrial and ventricular loading conditions, leading to a higher NP level. Unfortunately, precise data about the LA dimension, diastolic function (i.e., assessed with E/e’ ratio), and systolic pulmonary pressure were not provided in the two trial populations. Furthermore, the two studies did not obtain any information about overtime changes in NPs levels after the transcatheter MV repair procedure. Therefore, their potential relationship to echocardiographic parameters and to clinical outcomes could not be assessed in these trials.

## 5. NPs and Other Biomarkers

The most plausible pathophysiological explanation for the better association of NPs with pre- and post-procedural outcomes may be their optimal capacity to reflect cardiac performance in different hemodynamic conditions. Of note, lack of a sufficient number of studies using ANP as a marker in the management of MR does not currently allow to make a robust comparison with BNP.

Several efforts have been made over the last few years to investigate the role of other biomarkers in the MR condition, and have been previously reviewed [49]. More recently, other biomarkers have been investigated with some interesting insights.

The neutrophil gelatinase-associated lipocalin (NGAL) and cystatin C, both markers of functional and structural kidney damage, were shown to predict mortality in high-risk patients undergoing percutaneous MV repair [50]. However, they had low accuracy, probably due to a lack of relationship with the hemodynamic balance [35]. Similarly, the highly sensitive C-reactive protein (hsCRP), a biomarker able to improve risk prediction for cardiovascular diseases, showed low performance in the prognostic assessment of the MR patients [35]. On the other hand, the pre-operative level of soluble ST-2, a member of the interleukin-1 receptor family previously described as a stronger biomarker of myocardial stretch in HF, was correlated with LV function and structure after MV repair, thus providing complementary prognostic information to NT-proBNP level [35]. The level of galectin-3, a well-established marker of LV fibrosis, has been associated with worse cardiovascular outcomes after MV repair [35]. Interestingly, low galectin-3 and ST2 plasma levels were predictors of therapeutic success in 210 patients treated with percutaneous MV repair (PMVR) [51]. Of note, a lower galectin-3 level was a predictor of MR improvement after cardiac resynchronization therapy (CRT) [52]. In addition, biomarkers reflecting inflammation (hsCRP, interleukin-6) and cardiac remodeling processes (matrix metalloproteinases (MMP-2 and MMP-9)) were associated with a higher risk of mortality following the procedure [53]. The highly sensitive troponin T showed strong prognostic power in predicting survival after transcatheter MV repair, with an accuracy comparable to that of a mid-regional proANP level [35].

Of note, some evidence of the role of other parameters in the outcome prediction of MR patients has been collected. For instance, it was found that abnormalities of the calcium-phosphate metabolism may influence the health-related quality of life in patients with severe MR [54]. Amelioration of oxidative stress and endothelial dysfunction may be indicators of successful MV repair in patients with MV prolapse [55].

Finally, one study reported the negative prognostic impact of pre-procedural anemia in patients who underwent PMVR with a higher baseline NT-proBNP level [56].

Based on the evidence collected so far, markers of mechanisms involved in the cardiac remodeling process may be considered as complementary prognostic tools to the NP level.

## 6. Conclusions

The management of MR still represents a real challenge for physicians, since the assessment of symptoms is difficult and the recommended diagnostic exams, such as the echocardiogram, are often not accurately performed. It is well-known that NP secretion occurs in the presence of atrial and ventricular stretch, as a consequence of pressure and volume overload, and that NPs are independent predictors of mortality and morbidity in patients with severe MR. Due to their feasible measurement and to their significant association with echocardiographic parameters of LV dysfunction and of impaired filling, these biomarkers may represent an important tool to identify patients at elevated risk of adverse clinical outcomes, in which early surgical or percutaneous intervention should be considered (Figure 1 and Figure 2).

Moreover, the NP level has also been documented to be a powerful independent predictor of reduced cardiac event-free survival in patients treated with surgical or transcatheter valve repair or replacement, thus representing potential instruments to significantly improve the evaluation of a short- and long-term prognosis after these procedures.

Although some controversies still exist, the majority of findings discussed in our review article are encouraging, and indicate NPs as potentially useful biomarkers for the clinical management of MR. Further larger studies are needed to solve key issues, that is, to better define which biomarker should preferably be used among NPs, which rest and eventually stress cut-off levels could clearly identify high-risk patients, and which degree of variation between serial measurements may have the best clinical accuracy. It is hoped that these future studies will allow the introduction of NP levels into the guidelines for the management of MR.

## Figures and Tables

**Figure 1 jcm-09-01348-f001:**
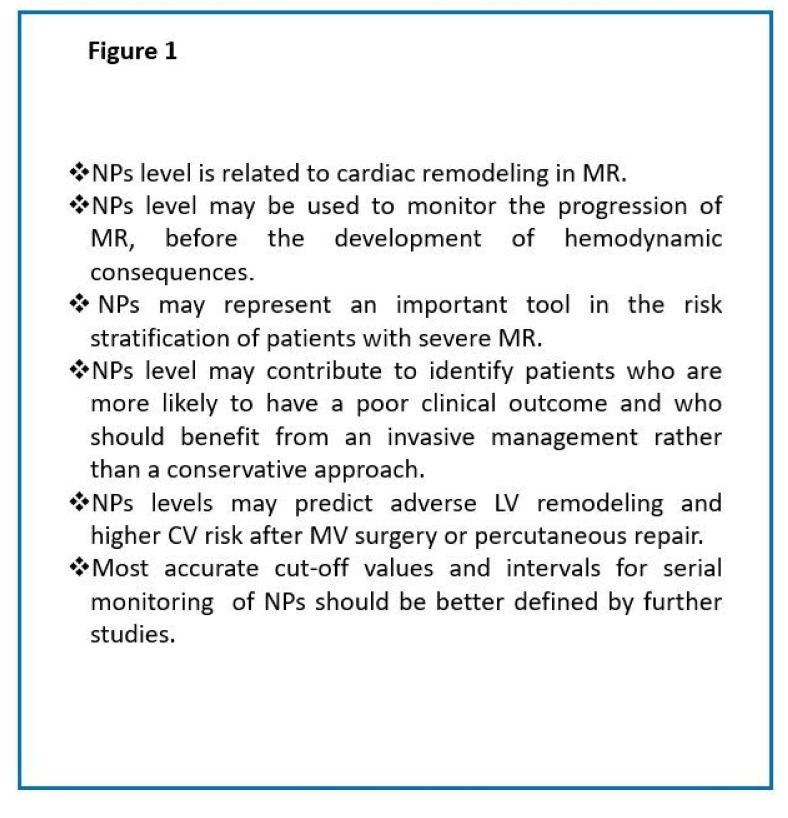
Summary of the main topics discussed in this review. Abbreviation legends: CV = cardiovascular; MR = mitral regurgitation; MV = mitral valve; NPs = natriuretic peptides.

**Figure 2 jcm-09-01348-f002:**
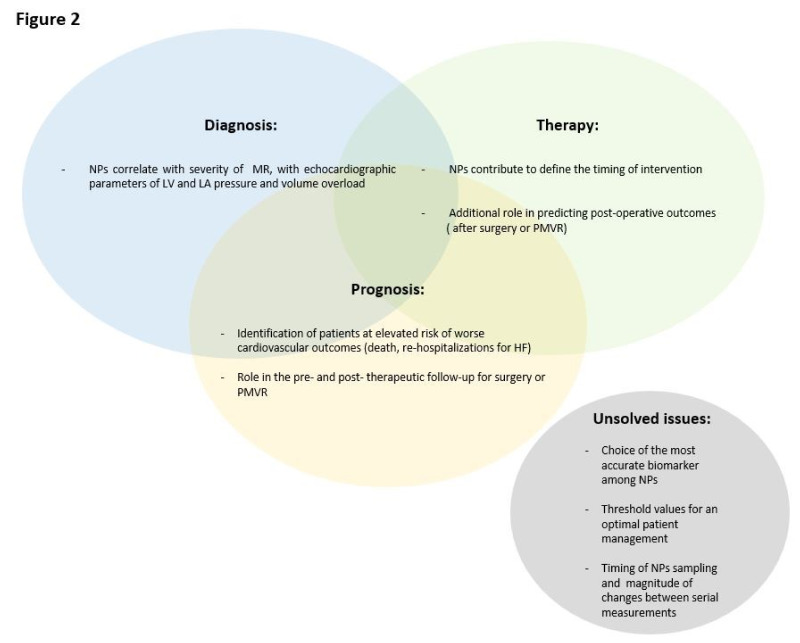
Main clinical implications of NPs in MR. Abbreviation legends: HF = Heart failure; LA = left atrium; LV = left ventricle; NPs = natriuretic peptides; PMVR = percutaneous mitral valve repair.

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
