# Peer review of "Functional Role of Natriuretic Peptides in Risk Assessment and Prognosis of Patients with Mitral Regurgitation"

_jcm, 2020, doi:10.3390/jcm9051348_

Round 1

Reviewer 1 Report

In the present manuscript, Dr. Giovanna Gallo, et al demonstrated the impact of natriuretic peptides on MR. This review was focused on not only surgical but also transcatheter intervention. Now a days, we have to consider surgical risk (i.e. surgical risk score, frailty) and patient condition (i.e. NPs), and select ideal patients for intervention. NPs are one of significantly useful for decision of timing for intervention. The present review is educational, and readers could reconsider the importance of biomarkers.

Readers of the Journal of Clinical Medicine will take interest in the present manuscript. However, there are several issues to consider:

Major issue

1)

From several previous reports, natriuretic peptides, especially BNP, was one of significant predictor for poor outcomes on many pathologies. As you mention it, This could clarify the risk for outcomes and good parameter for follow-up.

Compared between the previous two big data, MitraClip therapy should be performed for not so large LV and severe MR. By measurements of NPs, it was difficult to decide “too late” for intervention. Regardless of large LV, NPs are not so high if it was hemodynamically compensated. As I mention before, it means too hard that “NPs contribute to define the timing of intervention”. Should intervention be performed earlier if NPs are high? Please mention this more.

Author Response

We thank this Reviewer for his/her positive comments on our review article.

Specific comment:

"Compared between the previous two big data, MitraClip therapy should be performed for not so large LV and severe MR. By measurements of NPs, it was difficult to decide “too late” for intervention. Regardless of large LV, NPs are not so high if it was hemodynamically compensated. As I mention before, it means too hard that “NPs contribute to define the timing of intervention”. Should intervention be performed earlier if NPs are high? Please mention this more."

The Reviewer raises a good point. In order to accomplish it, we introduced a new sentence in pg.4, lines 161-163: “…On the other hand, it should be pointed out that, in the presence of a good hemodynamic compensation with normal NPs level, regardless of LV dimension, the need of an interventional strategy may be missed.”

Reviewer 2 Report

The review is very interesting. Overall the topic of the review is of relevance for the scientific community and I think worth being published. However, I have some concerns:

The Authors should incorporate a pictorial or cartoon representation of the topics discussed in the Review in order to facilitate the comprehension and increase the overall impact of the manuscript.

Please change the title as follows:"Functional role of natriuretic peptides in risk assessment 2 and prognosis of patients with mitral regurgitation".

The following recent reports should be mentioned:

Flint N, Raschpichler M, Rader F, Shmueli H, Siegel RJ. Asymptomatic Degenerative Mitral Regurgitation. JAMA Cardiol. 2020

Mozenska O, Bil J, Segiet A, Kosior DA. The influence of calcium-phosphate metabolism abnormalities on the quality of life in patients with hemodynamically significant mitral regurgitation. BMC Cardiovasc Disord. 2019 May 16;19(1):116.

Porro B, Songia P, Myasoedova VA, Valerio V, Moschetta D, Gripari P, Fusini L, Cavallotti L, Canzano P, Turnu L, Alamanni F, Camera M, Cavalca V, Poggio P. Endothelial Dysfunction in Patients with Severe Mitral Regurgitation. J Clin Med. 2019 Jun 12;8(6).

Kaneko H, Neuss M, Okamoto M, Weissenborn J, Butter C. Impact of Preprocedural Anemia on Outcomes ofPatients With Mitral Regurgitation Who UnderwentMitraClip Implantation. Am J Cardiol. 2018 Sep 1;122(5):859-865.

Lugnier C, Meyer A, Charloux A, Andrès E, Gény B, Talha S. The Endocrine Function of the Heart: Physiology and Involvements of Natriuretic Peptides and Cyclic Nucleotide Phosphodiesterases in Heart Failure. J Clin Med. 2019 Oct 21;8(10).

Beaudoin J, Singh JP, Szymonifka J, Zhou Q, Levine RA, Januzzi JL, Truong QA. Novel Heart Failure Biomarkers Predict Improvement of Mitral Regurgitation in Patients Receiving Cardiac Resynchronization Therapy-The BIOCRT Study. Can J Cardiol. 2016 Dec;32(12):1478-1484.

Author Response

We thank this Reviewer for his/her positive comments on our article. We took in consideration his/her suggestions and modified the manuscript accordingly.

A point-by-point reply to the specific comments follows.

  1. As suggested, we included a new Figure (Figure 1) summarizing the main topics afforded in the review article.
  2. We modified the title of the article as suggested by the Reviewer.
  3. As recommended by the Reviewer, we included all indicated recent reports (see References highlighted in yellow in the revised version of the Manuscript).

Round 2

Reviewer 1 Report

It was well revised. 

Thanks.